# Soilless Culture Applications for Early Development of Soybean Crop (*Glycine max* L. Merr)

**Rosnani Abd Ghani** *, **Suhana Omar**, **Márton Jolánkai**, **Ákos Tarnawa**, **Zoltán Kende**, **Noriza Khalid**, **Csaba Gyuricza and Mária Katalin Kassai**

Institute of Agronomy, Hungarian University of Agriculture and Life Sciences, Páter Károly u. 1, H-2100 Gödöllő, Hungary; hana@mardi.gov.my (S.O.); jolankai.marton@uni-mate.hu (M.J.); tarnawa.akos@uni-mate.hu (Á.T.); kende.zoltan@uni-mate.hu (Z.K.); norizahockey@gmail.com (N.K.); gyuricza.csaba@uni-mate.hu (C.G.); kassai.maria.katalin@uni-mate.hu (M.K.K.)
* Correspondence: rosnani@mardi.gov.my

**Abstract:** Soilless cultivation systems in a controlled environment are increasingly being used due to several global issues such as climate change, pest and disease problems, declining soil fertility quality and limited agricultural land. Soybean is one of the potential crops that can be grown using the soilless planting system in a controlled environment. Therefore, a study was conducted to investigate the effect of nutrient concentrations on the early development of two soybean varieties. Four different nutrient concentrations (0%, 50%, 100% and 150%) were applied, and two soybean varieties (Martina and Johanna) were tested. This study was designed in a split-plot experimental design where the nutrient concentration was the main plot and the variety was the sub-plot. The data record was started after fertilization. The plant growth data were collected for five weeks. All the data were analyzed statistically using SPSS V.23 software. The results of the study found that the nutrient concentration affected the number of leaves and leaf area. The number of leaves was higher in the treatment using 100% and 150% nutrients. Meanwhile, the leaf area increased with increasing nutrient concentration from 0% to 150%. The nutrient concentration then interacted with the variety in influencing the plant height. The plant height of the Martina variety was higher than the Johanna variety when 0% and 100% nutrients were applied. The variety also affected the leaf area and interacted with the number of weeks in affecting the SPAD readings and number of leaves. Thus, the two tested varieties (Johanna and Martina) have distinct early growth patterns that differ from one another as affected by the nutrient concentrations and plant age (number of weeks).

**Keywords:** soilless substrate; controlled environment; nutrient concentration; early development; soybean; Johanna variety; Martina variety; plant growth

## 1. Introduction

Soilless crop cultivation, especially in a controlled environment, is one of the alternatives for crop production in this era. It is an important alternative due to several global issues in the agricultural industry related to food production. Among the issues are climate change, the threat of pests and diseases, a decline in soil fertility as well as the area of land for agricultural activities that are decreasing and limited. Furthermore, crop production in a controlled environment can maintain the quality of the crops to supply yield or raw material continuously, including during the off-season [1,2].

For crop production in a controlled environment, regardless of conventional or new cultivation technologies, it is necessary to emphasize some important things such as planting systems and equipment, environmental conditions (temperature, humidity, lighting, ventilation, carbon dioxide ($CO_2$)), soil or type of growing medium, nutrients and irrigation systems [3,4]. For soilless crop cultivation in a controlled environment, the selection of the appropriate soilless culture growing medium is necessary to guarantee good plant growth

and yield production. The selection depends on the type of crop, cultivation system, the pH of the irrigation water, cost, and the shelf life of the system equipment [5]. The most important things that the soilless culture needs to provide are oxygen, water, nutrients and support for the plant roots as the functions of soil [5]. Among the recommended and ideal soilless cultures besides water culture (hydroponics) are organic cultures, such as peat, bark, coir, rice hulls, etc., and inorganic substrate cultures such as gravel, sand, perlite, rockwool, volcanic stones, etc. [3]. In addition, inorganic substrate cultures such as expanded clay aggregate or clay mineral aggregates are widely used because their spherical shape and porosity help provide a good balance between oxygen and water so that plant roots are not overly dry or drowned. The clay mineral aggregate substrate releases almost no nutrients and has a neutral pH 7.0 [3]. It is reported to be a beneficial component of pine bark and peat-based soilless substrates [6]. The soilless production system has been used worldwide and is currently relied on heavily for greenhouse vegetable production in Europe, the United States, the Middle East, Japan and Canada [7]. Various plants are suitable for growing using soilless culture in a controlled environment. Soybean is one of the potential crops that can be cultivated under soilless conditions because it can utilize nitrogen derived from biological nitrogen ($N_2$) fixation and decrease the need for mineral nitrogen fertilization [8]. The reaction of the soybean plant to any situation faced such as the application of fertilizers, pesticides or other chemicals is very dependent on its stage of development [9]. The early stage of plant growth is critical because it affects the reproductive phase of a plant [10]. Other planting factors such as planting date, variety, location and weather will affect growth including the amount of stem and leaf tissue when flowering begins [9]. Therefore, it is crucial to study the influence of these planting factors such as fertilization and variety on the early growth of soybean using soilless culture in a controlled environment which is lacking in peer-reviewed research.

Crop plants including soybeans need nutrients for growth and yield production. Nutrients are the key factors for crop production and can give either a positive or negative effect on growth and yield depending on the amount, plant growth stage, combination and balance [11]. For example, the use of nutrients by plants decreases slowly after the plants reach the peak growth stage until maturity [12]. Macro- and micro-nutrients handle different morphological and physiological functions in plants [13]. Under nutrient-deficiency conditions, plants use specific mechanisms to alleviate stress which will cause an increase in the uptake of nutrients. Meanwhile, excessive nutrient concentration will interfere with the uptake and consumption of other nutrients [14]. Therefore, the efficient use of fertilizer by plants is essential because it will increase the growth rate and crop yield. Based on previous studies, a comparison of hydroponics systems using the nutrient film technique (NFT) method and substrate culture using rockwool was undertaken, in which two sources of nitrogen fertilizers (nitrate fertilizers and urea) were also tested together with the two techniques on soybean growth [15]. The results of the study found that the number of leaves and leaf area of the soybeans increased throughout the first eight weeks of planting, which reached a maximum in the eighth week regardless of the crop system and the type of nitrogen fertilizer. Additionally, the use of rockwool increased the plant height of the soybeans.

There was also a study using different soilless cultures (rockwool, husk charcoal and cocopeat) in the paranet house system that was performed on two varieties of lettuce. The results from the study found that there was a significant increase in the height of the plant from 9, 16, 23 and 30 days after planting [16]. Based on a study conducted by Hata and Futamura [17], the use of nutrient solution (Enshi nutrient solution) with different concentrations had a significant effect on soybean growth. They found that hydroponically grown soybeans with substrate silica sand + Rhizobium inoculant influenced the stem length (peak with 25% dilution) and the number of trifoliate leaves which both increased with increasing concentration from 0% to 50% using nitrate as a source of nitrogen. They also found that the plant leaves treated with concentrations of 0–25% were significantly less green than the control plants (complete nutrient solution without inoculant) but the green leaf color intensity was the same for the plants treated with 50% and the control plants.

Meanwhile, a study on three varieties of faba beans using commercial nutrient solution (Cooper, 1979) with several concentrations (0, 25%, 100%, 300%) found that there was a significant increase in the number of leaves with the treatment of 0% and 25% nutrient concentrations [18]. They also found that the leaf area significantly increased with an increase in the concentration of the nutrient solution for all three varieties of faba bean.

In addition, a study related to different nutrient concentrations by Nurul Aini [19] found that nutrient concentrations of 100% (electrical conductivity (EC) 1.8 dS m$^{-1}$) and 75% (EC 1.4 dS m$^{-1}$) did not show differences in the leaf thickness, leaf area and lettuce yield at 28 days after planting (DAP) and during harvesting (42 DAP). Meanwhile, a nutrient concentration with 50% (EC 0.9 dS m$^{-1}$) decreased the leaf thickness, leaf area and total yield at both 28 DAP and 42 DAP. Similar results were reported by Walters [20] when the nutrient solutions with different EC did not affect plant growth including the plant height and soil plant analysis development (SPAD) reading for three types of basil (sweet basil, lemon basil and holy basil). There was also a study on tomato seedlings grown in glasshouse conditions using a commercial growing medium (Barocer, Seoul-bio Co., Eumseong, Republic of Korea) and a deep-flow culture container system. The results of the study found that the number of leaves, fresh weight and dry weight of tomato seedlings 10 days after transplanting was the highest at a nutrient concentration of fivefold (113.1 mg L$^{-1}$ ammonium nitrogen (NH$_4$-N), 507.2 mg L$^{-1}$ nitrate ntrogen (NO$_3$-N), 98.5 mg L$^{-1}$ phosphorus (P), 938.2 mg L$^{-1}$ potassium (K), 245.2 mg L$^{-1}$ calcium (Ca), 123.1 mg L$^{-1}$ magnesium (Mg)) and potential hydrogen (pH) 6, but not significant differences at other nutrient concentrations and pH [21].

Based on the previous studies above, it can be concluded that nutrient concentration and the use of soilless culture have positive and negative effects on the early growth development of plants in a controlled environment which also depends on the crop type and age of the plant. Although previous studies found that different basil varieties did not show a significant effect due to the influence of nutrient solution, there is less information available for soybean varieties, especially cultivation using soilless culture in a controlled environment. Thus, a study was conducted to investigate the effects of nutrient concentration on the early development of two soybean varieties. The results of the study provide the basis for the development of a good quality and pesticide-free soybean production system.

## 2. Materials and Methods

### 2.1. Location, Planting System and Growth Conditions

A soybean pot study was conducted under a controlled environment in a 4 m × 1.8 m plant growth chamber. The growth chamber was placed at the experimental site of the Institute of Agronomy, the Hungarian University of Agriculture and Life Sciences (MATE) in Gödöllő, Hungary with coordinates 47°46′ N, 19°21′ E and 242 m above sea level. This study was conducted from January 2022 until July 2022.

The soilless cultivation system was used in this experiment using a substrate culture. The growth chamber was equipped with air conditioning, fluorescent lights, fans and sets of planting systems. An air conditioner was installed to provide a suitable ambient temperature in the chamber. Meanwhile, fluorescent lamps with red and blue light (58 watts) were installed according to the amount of light required to provide enough light for plant growth. Two fluorescent lights were installed for each main plot (nutrient concentration treatments). Meanwhile, the planting systems were equipped with pots, irrigation systems, tanks, water pumps and timers.

The environment in the growth chamber was set up with the same condition throughout the growing season. Daylight with a light intensity of 950 lux was set up for 16 h and nighttime was set up for 8 h with the temperature of 22 °C during the daylight and 16 °C during the nighttime. The humidity in the growth chamber was around 40–60%. Plants were planted using planting sets equipped with pots 19 cm diameter and 22 cm high (10 L capacity), a drip irrigation system, nutrient solution tanks with the size of 117 cm × 60 cm and water pumps with a capacity of 1000 L/h (Newa Maxi IP68, Loreggia

PD, Italy; 220–240 V, 13 W). A substrate culture of expanded clay aggregate was used as a growing medium. Before planting, a drip irrigation system was installed for each main plot with a water pump placed for each nutrient solution tank.

## 2.2. Treatments and Experimental Design

The study was a factorial experiment with nutrient concentration as the first factor, and variety as the second factor. It was designed according to a split-plot experimental design with three replications where nutrient concentration was the main plot and variety was the sub-plot. The first factor consisted of four different nutrient concentrations which were 0% (control), 50%, 100% (full plant requirement) and 150%. A combination of Dutch Formula (Advance Hydroponics of Holland) liquid fertilizers were used in this study which were fertilizer formulation 1 (Grow), fertilizer formulation 2 (Bloom) and fertilizer formulation 3 (Micro). The nutrient content in each formulation is shown in Table 1. Meanwhile, the nutrient content in each nutrient concentration treatment is shown in Table 2. The recommended nutrient supply which was according to the full plant requirement for all the three formulation fertilizers that needed to be diluted in 100 L of water is shown in Table 3. All the fertilizers were supplied based on two recommended rates according to the growth stages. The first rate was supplied at vegetative stage 1 (V1) and vegetative 2 (V2) supplied in Week 1 and Week 2, respectively. The second rate was for vegetative stage 3 (V3), vegetative stage 4 (V4) and vegetative stage 5 (V5) that were supplied, respectively, in Weeks 3, 4 and 5. In this study, the amount of water that was filled into the tank for each treatment was 25 L. The amount of fertilizer that needed to be diluted in 25 L of water for each treatment is shown in Table 4. The nutrient solution was applied after 10 days of germination for all the treatments.

**Table 1.** Nutrient content (%) for three formulations of Dutch Formula liquid fertilizer.

| Nutrient | Nutrient Content (%) | | |
|---|---|---|---|
| | Formulation 1 (Grow) | Formulation 2 (Bloom) | Formulation 3 (Micro) |
| Nitrate ($NO_3$) | 1.8 | 0.3 | 4.5 |
| Ammonium ($NH_4$) | 0.6 | 0.4 | 0 |
| Phosphorus pentoxide ($P_2O_5$) | 4.4 | 5.7 | 0 |
| Potassium oxide ($K_2O$) | 7.4 | 5.3 | 3.0 |
| Magnesium oxide (MgO) | 0.8 | 2.1 | 0 |
| Sulfur trioxide ($SO_3$) | 2.2 | 5.6 | 0 |
| Calcium oxide (CaO) | 0 | 0 | 6.0 |
| Boron (B) | 0 | 0 | 0.015 |
| Molybdenum (Mo) | 0 | 0 | 0.01 |
| Copper (Cu) | 0 | 0 | 0.006 |
| Manganese (Mn) | 0 | 0 | 0.04 |
| Zinc (Zn) | 0 | 0 | 0.02 |
| Iron (Fe) | 0 | 0 | 0.15 |

**Table 2.** Nutrient content (%) of each nutrient concentrations.

| Nutrient | Nutrient Content (%) | | | |
|---|---|---|---|---|
| | 0% | 50% | 100% | 150% |
| Nitrate ($NO_3$) | 0 | 0.83 | 1.65 | 2.48 |
| Ammonium ($NH_4$) | 0 | 0.13 | 0.25 | 1.52 |
| Phosphorus pentoxide ($P_2O_5$) | 0 | 1.27 | 2.53 | 3.80 |
| Potassium oxide ($K_2O$) | 0 | 1.97 | 3.93 | 5.89 |
| Magnesium oxide (MgO) | 0 | 0.37 | 0.73 | 1.1 |
| Sulfur trioxide ($SO_3$) | 0 | 0.98 | 1.95 | 2.93 |
| Calcium oxide (CaO) | 0 | 0.75 | 1.5 | 2.25 |
| Boron (B) | 0 | 0.0019 | 0.004 | 0.006 |
| Molybdenum (Mo) | 0 | 0.0013 | 0.003 | 0.038 |
| Copper (Cu) | 0 | 0.008 | 0.015 | 0.023 |
| Manganese (Mn) | 0 | 0.005 | 0.01 | 0.015 |
| Zinc (Zn) | 0 | 0.003 | 0.005 | 0.008 |
| Iron (Fe) | 0 | 0.019 | 0.038 | 0.057 |

**Table 3.** The amount of Dutch Formula liquid fertilizer recommended to be diluted in 100 L of water for all three formulations according to plant growth stages.

| Ferilizer | V1 and V2 | V3, V4 and V5 |
| --- | --- | --- |
| | Total Fertilizer in mL | |
| Formulation 1 (Grow) | 75 | 150 |
| Formulation 2 (Bloom) | 37 | 75 |
| Formulation 3 (Micro) | 37 | 75 |

**Table 4.** The amount of Dutch Formula liquid fertilizer that was diluted in 25 L of water according nutrient concentration treatments and different vegetative stage.

| Fertilizer | V1 and V2 | | | | V3, V4 and V5 | | | |
| --- | --- | --- | --- | --- | --- | --- | --- | --- |
| | Total Fertilizer in mL | | | | | | | |
| | 0% | 50% | 100% | 150% | 0% | 50% | 100% | 150% |
| Formulation 1 (Grow) | 0 | 9.38 | 18.75 | 28.13 | 0 | 18.75 | 37.5 | 56.25 |
| Formulation 2 (Bloom) | 0 | 4.63 | 9.25 | 13.88 | 0 | 9.38 | 18.75 | 28.13 |
| Formulation 3 (Micro) | 0 | 4.63 | 9.25 | 13.88 | 0 | 9.38 | 18.75 | 28.13 |

The second factor consisted of two different soybean varieties, which were Martina and Johanna. Seeds for both soybean varieties with a high germination percentage of over 90% were used and planted directly into the pots. A total of 8 soybean seeds were planted for each pot. After 10 days of planting, only 6 healthy and uniform seedlings remained in each pot and the other two were removed. Therefore, there was one pot of 6 seedlings for each variety, nutrient concentration and replication.

*2.3. Crop Management and Measurements*

After the seeds were directly planted into the pots, they germinated for 10 days. During the germination period, irrigation water with no nutrients was automatically irrigated for 30 min three times a day. After 10 days, the nutrient solution was also supplied three times a day for 30 min per irrigation. The nutrient solution in the tanks was changed manually once a week to maintain the potential hydrogen (pH) and electrical conductivity (EC) at the appropriate level. The data measurements were collected a week after the nutrients were supplied. The data were recorded every week until the plants produced flowers, with 5 weeks of data measurement (Week 1, Week 2, Week 3, Week 4, Week 5). The growth data measured included the plant height, number of leaves, soil plant analysis development (SPAD) reading and leaf area. For the SPAD reading measurements, the green leaf colour intensity of fully expanded second trifoliate leaves was measured using a SPAD-502 chlorophyll meter (Minolta Camera, Osaka, Japan). Meanwhile, the leaf area was obtained using a non-descriptive method by directly measuring the maximum length and width of the leaves on the plant. The growth data were recorded for all the 6 plants in each pot. The average from the 6 plants were then analyzed statistically.

*2.4. Data Analysis*

All the recorded data were statistically analyzed using IBM SPSS V.23 software (SPSS Inc., Chicago, IL, USA). The results presented are the mean value data for the main effect of nutrient concentration, variety, weeks and the mean value of interaction effects. Statistical differences at $p < 0.05$ between the analyzed parameters were obtained using a three-way analysis of variance (ANOVA) followed by the least significant difference (LSD) test at $p < 0.05$.

## 3. Results

### *3.1. Plant Height*

The results of the studies showed that there were no significant differences between the nutrient concentration treatments on the plant height at the early development stage of soybeans grown using soilless culture under the controlled environment (Table 5). The study also showed (Table 5) that there was no significant difference between the soybean varieties on the plant height. However, the week number significantly affected the plant height at $p < 0.05$. As seen in Figure 1, the plant height increased every week until the fifth week.

**Table 5.** Analysis of variance (ANOVA) for plant height of soybeans as affected by nutrient concentration, variety and week number.

| Source | Sum of Squares | df | Mean Square | F | Sig. |
|---|---|---|---|---|---|
| Nutrient concentration (N) | 43.6 | 3 | 14.53 | 2.10 | 0.11 |
| Variety (V) | 15.48 | 1 | 15.48 | 2.23 | 0.14 |
| Week (W) | 1609.26 | 4 | 402.32 | 58.04 | 0.00 |
| N × V | 127.52 | 3 | 42.51 | 6.13 | 0.00 |
| N × W | 24.91 | 12 | 2.08 | 0.30 | 0.99 |
| V × W | 14.39 | 4 | 3.60 | 0.52 | 0.72 |
| N × V × W | 32.43 | 12 | 2.70 | 0.39 | 0.96 |
| Error | 554.51 | 80 | 6.93 | | |
| Total | 2422.13 | 119 | | | |

df: Degree of freedom; F: F statistic; Sig.: Significance; Significance level = $p < 0.05$.

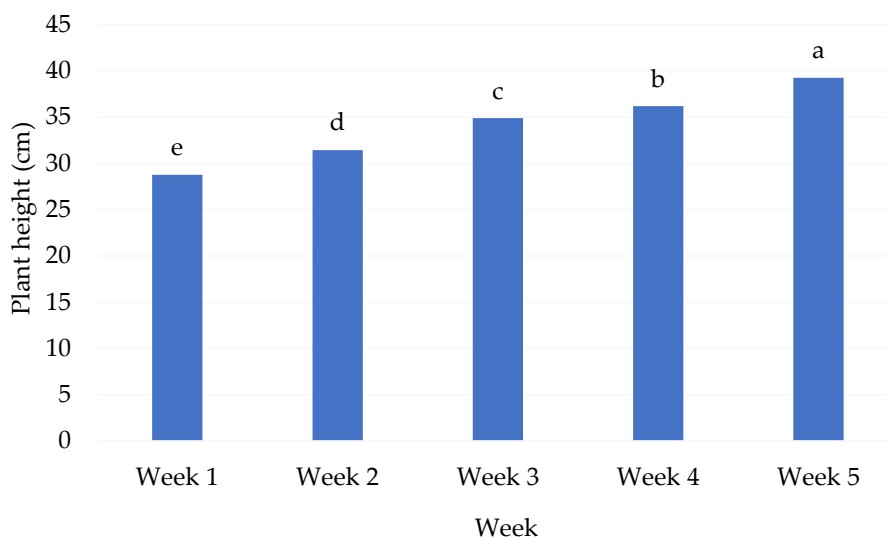

**Figure 1.** Effect of week number on plant height at early growth of soybeans grown using soilless culture. Means with the same letter are not significantly different from one another by LSD at $p < 0.05$.

There was also an interaction effect between the nutrient concentration and variety on the soybean plant height (Table 5). The interaction effect is shown in Figure 2. The plant height of the Martina variety was higher than the Johanna variety when no nutrient (0%) was applied. When the nutrient was applied up to 50% of the complete plant requirement, the Martina and Johanna varieties showed almost the same height, with a value of 34.17 cm and 34.21 cm, respectively. However, the plant height of the Martina variety was higher (36 cm) than the Johanna variety (32.35 cm) when the nutrient was supplied as much as 100% concentration. The Johanna variety was then higher in the plant height (35.88 cm) compared to the Martina variety (33.86 cm) at the nutrient concentration of 150%.

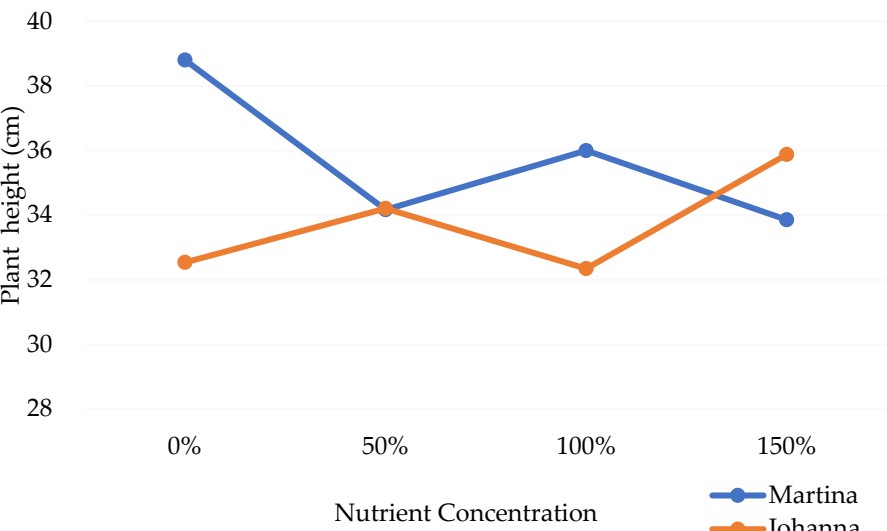

**Figure 2.** Interaction effect of nutrient concentration and variety on plant height at early growth of soybeans planted under soilless condition.

### 3.2. Number of Leaves

Meanwhile, the three main effects of nutrient concentration, variety and week number had a significant impact on the average number of leaves (Table 6). However, the number of leaves of the soybean plants with treatment without nutrient (0%) and with 50% nutrient were not significantly different at $p < 0.05$; the number of fully expanded leaves for both treatments were found to have seven leaves (Figure 3). They were the lowest and very significant at $p < 0.05$ compared with the number of leaves at 100% and 150%. Soybeans treated with 100% and 150% nutrient concentrations produced nine leaves, two more than the plants treated with 0% and 50% nutrients. The effect of the variety on the number of soybean leaves at the early growth stage showed that both varieties of Martina and Johanna had a significant difference at $p < 0.05$. The number of leaves for the Johanna variety was more than the number of leaves for the Martina variety which had 9 and 8 leaves, respectively. The main effect of the week number was an increasing trend of the number of leaves, which was similar to the growing trend for plant height. This is shown in Figure 4, in which the number of leaves had a significant difference at $p < 0.05$ with the increasing of the week number.

**Table 6.** Analysis of variance (ANOVA) for number of leaves of soybeans as affected by nutrient concentration, variety and week number.

| Source | Sum of Squares | df | Mean Square | F | Sig. |
|---|---|---|---|---|---|
| Nutrient concentration (N) | 45.76 | 3 | 15.25 | 21.28 | 0.00 |
| Variety (V) | 3.01 | 1 | 3.01 | 4.20 | 0.04 |
| Week (W) | 1085.00 | 4 | 271.25 | 378.49 | 0.00 |
| N × V | 4.56 | 3 | 1.52 | 2.12 | 0.10 |
| N × W | 14.87 | 12 | 1.24 | 1.73 | 0.08 |
| V × W | 13.87 | 4 | 3.47 | 4.84 | 0.00 |
| N × V × W | 6.40 | 12 | 0.53 | 0.74 | 0.70 |
| Error | 57.33 | 80 | 0.72 | | |
| Total | 1230.79 | 119 | | | |

df: Degree of freedom; F: F statistic; Sig.: Significance; Significance level = $p < 0.05$.

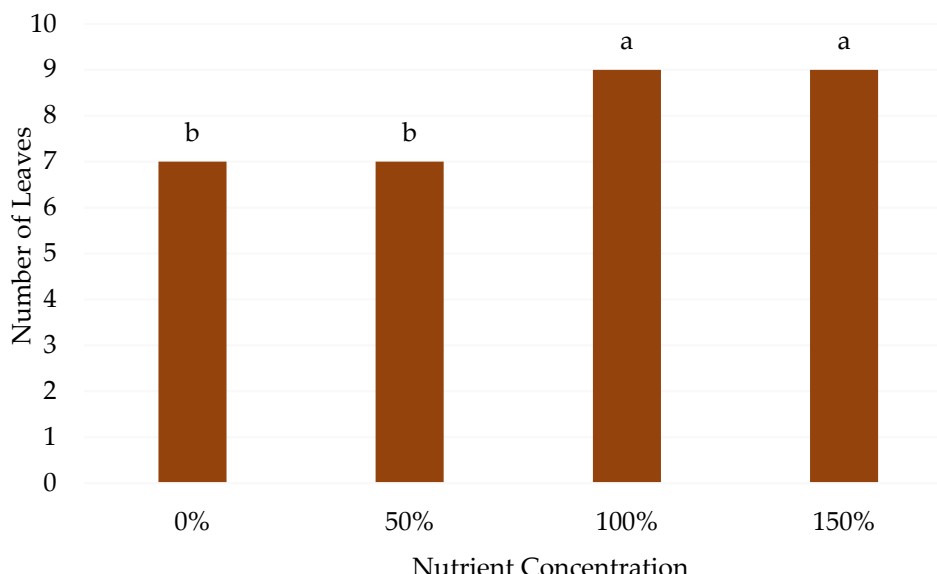

**Figure 3.** Effect of nutrient concentration on average number of leaves at early growth of soybean planted under soilless culture conditions. Means with the same letter are not significantly different from one another by LSD at $p < 0.05$.

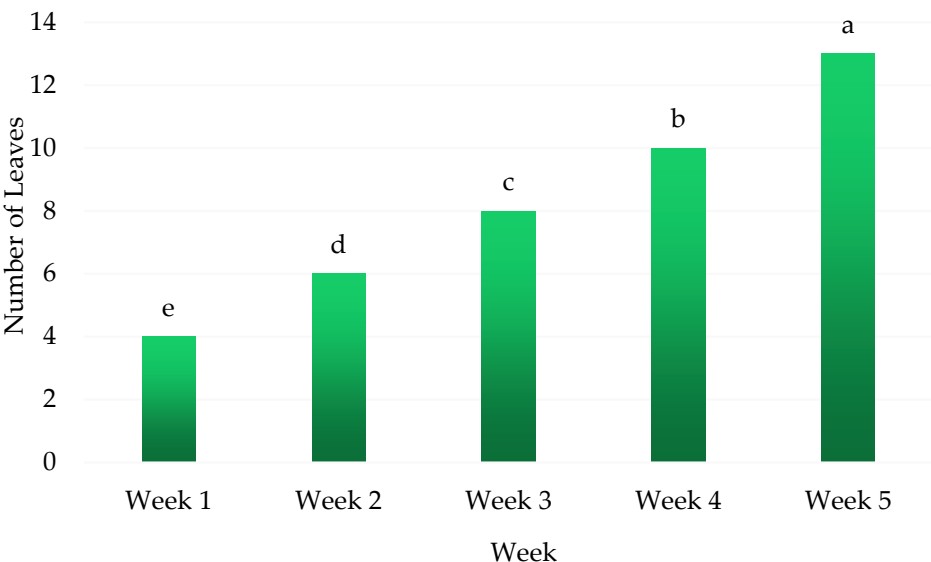

**Figure 4.** Effect of week number on number of leaves at early growth of soybeans planted under soilless culture conditions. Means with the same letter are not significantly different from one another by LSD at $p < 0.05$.

There was also a significant interaction between the variety and week on the number of leaves. The number of leaves for both the varieties increased with an increasing number of weeks (Figure 5). In Week 1, the number of leaves for the Johanna variety was higher than the Martina variety with five and three leaves, respectively. Meanwhile, the number of leaves for both varieties was six leaves in the second week and eight leaves in the third week. However, the Johanna variety increased to 10 leaves, and exceeded the number of leaves for the Martina variety (9 leaves) in Week 4. The number of leaves for the Martina variety then continued to increase in Week 5 and reached higher than the Johanna variety which was 13 leaves for Martina and 12 leaves for Johanna.

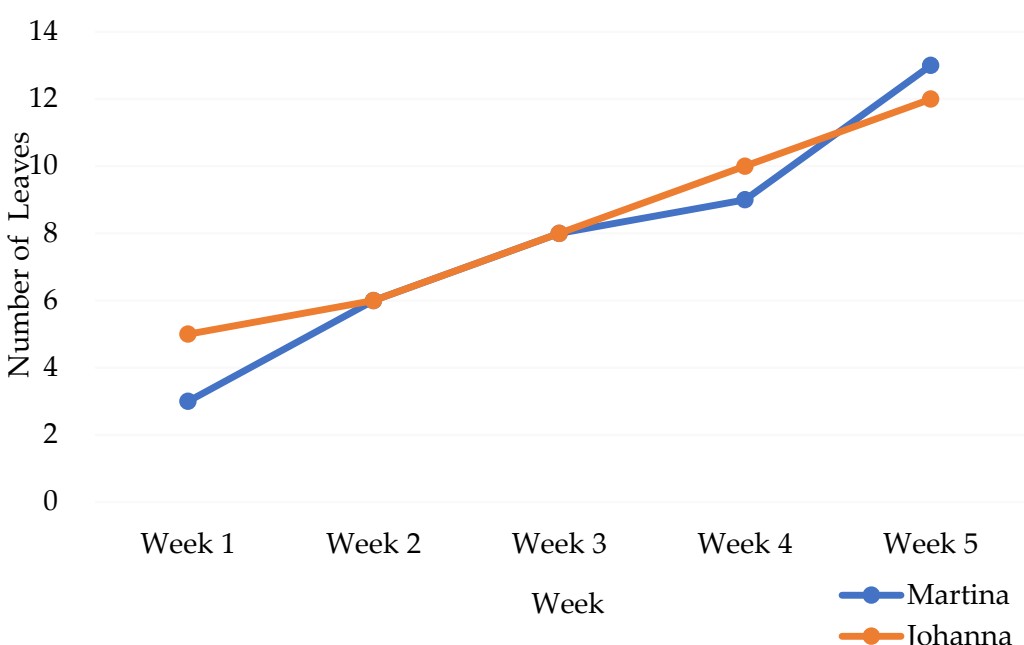

**Figure 5.** Interaction effect of variety and week number on number of leaves at early growth of soybean planted under soilless condition.

### 3.3. SPAD Reading

There were no significant differences for the main effect of nutrient concentration and variety for the SPAD reading of soilless culture soybeans at early development planted in a controlled environment (Table 7). However, the main effect of the week number was significant ($p < 0.05$) on the SPAD reading. The SPAD reading, which shows the chlorophyll content and green leaf color intensity, gave the lowest reading in Week 1 with a value of 33.92 and was very significant ($p < 0.05$) with Week 2, Week 3, Week 4 and Week 5 (Figure 6). The SPAD reading increased significantly in Week 2, which gave a value of 36.31 and continued to increase to 38.43 in Week 3. In Week 3, the results of the study found that the SPAD reading value was the highest and highly significant ($p < 0.05$) with the value from the two weeks before (Week 1 and Week 2) and with the value from the two weeks after (Week 4 and Week 5). The SPAD reading started to show a decrease of 0.96 from 38.43 in Week 3 to 37.47 in Week 4, and continued to decrease through Week 5 (Figure 6).

**Table 7.** Analysis of variance (ANOVA) for SPAD reading of soybeans as affected by nutrient concentration, variety and week number.

| Source | Sum of Squares | df | Mean Square | F | Sig. |
|---|---|---|---|---|---|
| Nutrient concentration (N) | 14.38 | 3 | 4.79 | 2.57 | 0.60 |
| Variety (V) | 3.23 | 1 | 3.23 | 1.73 | 0.19 |
| Week (W) | 284.24 | 4 | 71.06 | 38.08 | 0.00 |
| N × V | 13.13 | 3 | 4.38 | 2.35 | 0.08 |
| N × W | 22.72 | 12 | 1.89 | 1.01 | 0.44 |
| V × W | 53.45 | 4 | 13.36 | 7.16 | 0.00 |
| N × V × W | 15.08 | 12 | 1.26 | 0.67 | 0.90 |
| Error | 149.30 | 80 | 1.87 | | |
| Total | 555.52 | 119 | | | |

df: Degree of freedom; F: F statistic; Sig.: Significance; Significance level = $p < 0.05$.

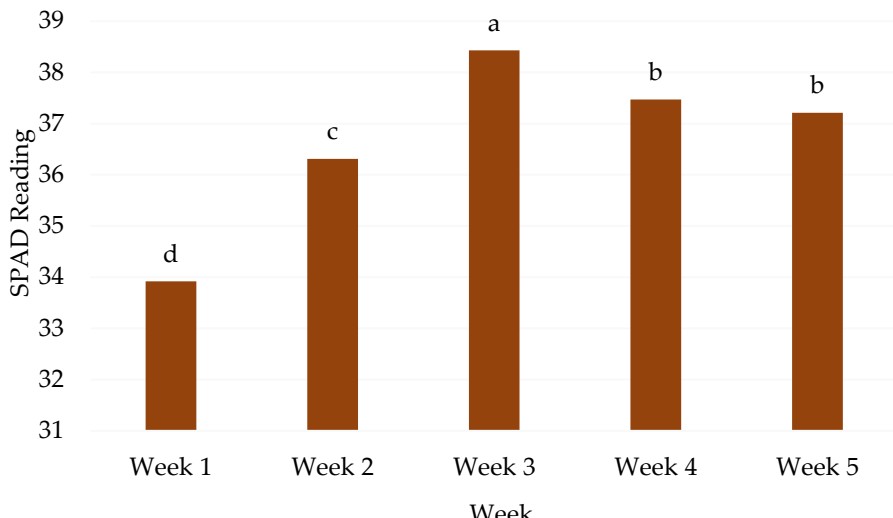

**Figure 6.** Effect of week number on SPAD reading at early growth of soybeans planted under soilless culture conditions. Means with the same letter are not significantly different from one another by LSD at *p* < 0.05.

A significant interaction also was found between the variety and week number on the SPAD reading (Table 7). The SPAD reading for both the varieties increased weekly until Week 3 (Figure 7). In Week 1, Week 2 and Week 3, the Johanna variety gave a higher SPAD reading than the Martina variety. In Week 1, the SPAD reading for the Johanna variety was 35.17 and the SPAD reading for the Martina variety was lower by 2.49 than the Johanna variety. The SPAD reading in Week 2 was 36.70 for the Johanna variety and for the Martina variety was 35.93. In Week 3, the SPAD reading was at a maximum value where Johanna and Martina, respectively, had a SPAD reading of 38.65 and 38.21. A decrease in the SPAD reading was found for both the varieties in Week 4 and Week 5. The Johanna variety decreased drastically and had a lower SPAD reading compared to the Martina variety in Week 4 and Week 5.

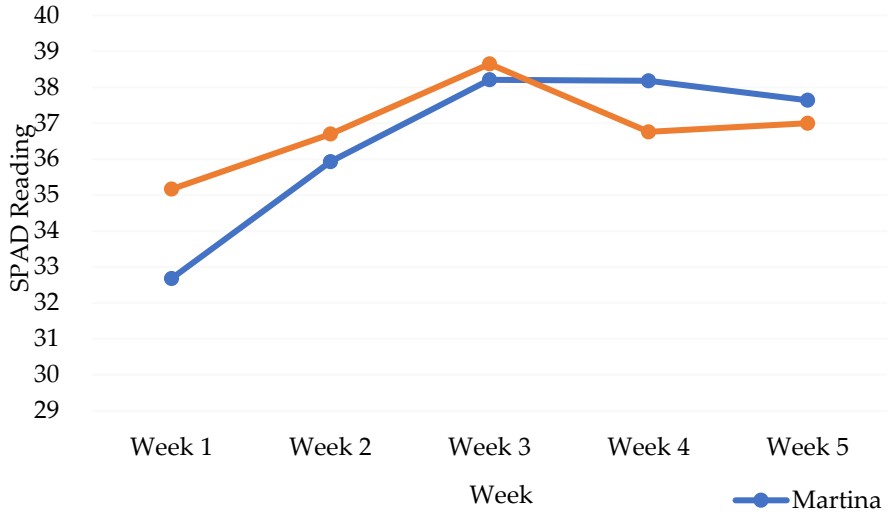

**Figure 7.** Interaction effect of variety and week on SPAD reading at early growth of soybeans planted under soilless conditions.

*3.4. Leaf Area*

All three factors that were tested, which were nutrient concentration, variety and week, showed significant effects on the leaf area (Table 8). However, there was no significant inter-

action between the treatments for all the tested factors. The effect of nutrient concentration on the leaf area showed an increase in the leaf size with increasing nutrient concentration (Figure 8). The leaf size was the largest and very significant in the early growth of soybeans that were supplied with nutrients that were more than the full plant requirement (150%). The leaf area of the soybean with a nutrient concentration of 150% was 37.40 cm$^2$ which was 1.98 higher than the size of the leaf with the full concentration of the plant requirement (100%) at 35.42 cm$^2$. Meanwhile, the plant without the nutrient treatment (0%) showed the smallest soybean leaf size which was 30.21 cm$^2$ and had a significant difference at $p < 0.05$ with other treatments. The leaf area for the 50% treatment was the second lowest with a leaf size of 32.37 cm$^2$. The main effect of variety on the leaf area of the soybeans showed that the Martina variety had a larger leaf area than the Johanna variety which were 36.66 cm$^2$ and 31.05 cm$^2$, respectively. Meanwhile, the result for the effect of the week number on the leaf area showed a significant difference at $p < 0.05$ between the weeks (Figure 9). The leaf area increased significantly from Week 1 until Week 3 and slightly decreased in Week 4. However, the leaf area in Week 4 was not significantly different from the leaf area in Week 3. The leaf area continued to decrease in Week 5 but did not show a significant difference from Week 4 and Week 2.

**Table 8.** Analysis of variance (ANOVA) for leaf area of soybeans as affected by nutrient concentration, variety and week number.

| Source | Sum of Squares | df | Mean Square | F | Sig. |
|---|---|---|---|---|---|
| Nutrient concentration (N) | 915.34 | 3 | 305.11 | 30.96 | 0.00 |
| Variety (V) | 943.94 | 1 | 943.94 | 95.80 | 0.00 |
| Week (W) | 644.78 | 4 | 161.20 | 16.36 | 0.00 |
| N × V | 62.77 | 3 | 20.92 | 2.12 | 0.10 |
| N × W | 37.56 | 12 | 3.13 | 0.32 | 0.98 |
| V × W | 8.34 | 4 | 2.09 | 0.21 | 0.93 |
| N × V × W | 27.01 | 12 | 2.25 | 0.23 | 1.00 |
| Error | 788.30 | 80 | 9.85 | | |
| Total | 3428.02 | 119 | | | |

df: Degree of freedom; F: F statistic; Sig.: Significance; Significance level = $p < 0.05$.

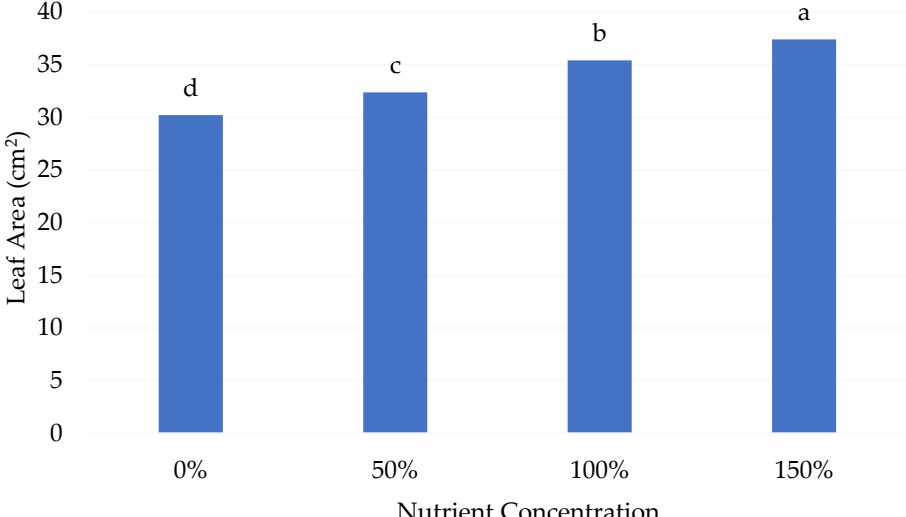

**Figure 8.** Effect of nutrient concentration on leaf area at early growth of soybeans planted under soilless culture conditions. Means with the same letter are not significantly different from one another by LSD at $p < 0.05$.

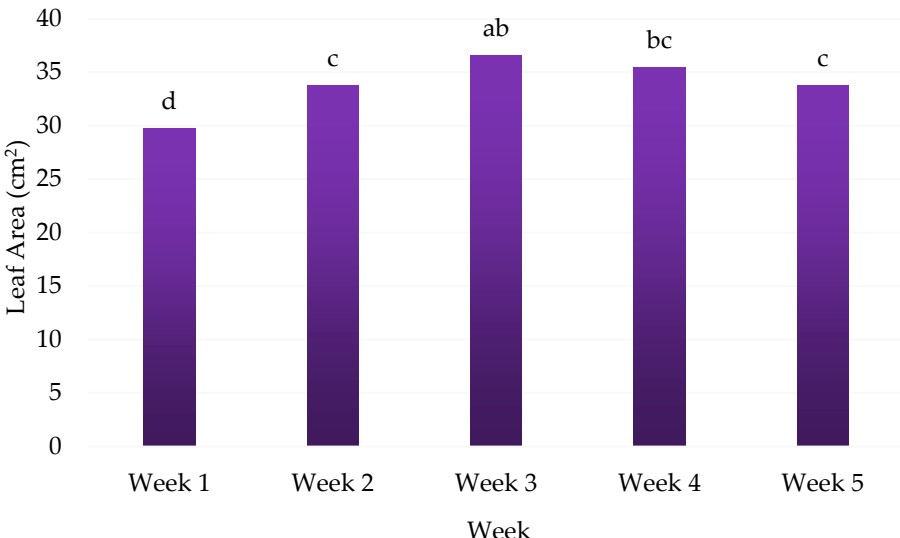

**Figure 9.** Effect of week number on leaf area at early growth of soybeans planted under soilless culture conditions. Means with the same letter are not significantly different from one another by LSD at $p < 0.05$.

## 4. Discussion

### 4.1. Week Number Affected Plant Height and Number of Leaves

The plant height and number of leaves were significantly increased with an increasing number of weeks at the early growth stage of soybeans grown in a controlled environment using a soilless culture (Figures 1 and 3). These are in line with studies conducted by Paradiso [15] on soybean plants and Purba [16] on lettuce. The explanation for this could be that every plant that is in excellent condition will continue to grow with increasing time. The plants were the most actively growing especially in the vegetative stage, including the increase in the plant height and number of leaves. For example, according to the early growth standard for soybeans from the stages of V2, V3, V4, V5 and V6, the plant height increments were around 15–20 cm, 18–23 cm, 23–25 cm, 25–31 cm and 31–36 cm, respectively [22]. However, the results from the study found that the plant height of the soybeans were higher (Figure 1) compared to the standard range. This difference is probably due to several factors including the difference in the cultivation system, cultivation condition and variety used.

### 4.2. Nutrient Concentration Affected Number of Leaves and Leaf Area

There was a significant increase only in the number of leaves from the nutrient concentration from 50% to 100% but there was no significant difference in the nutrient concentration from 0% to 50% (Figure 3). This was probably due to the soybean seeds being large and the storage compounds in the cotyledons providing the nutrients required for early plant growth [23]. This can explain why soybeans with 0% nutrients can also produce the same number of leaves as those with 50% nutrients even if no nutrient is supplied. However, the findings of this study differ from the findings of the survey by Hata and Futamura [17] where the number of trifoliate leaves increased with increasing nutrient concentration from 0% to 50%. This difference is probably due to the study by Hata and Futamura [17] using *Rhizobium* sp. inoculant for soybean cultivation by mixing rhizobia into the growing medium (silica sand). Therefore, apart from the nutrient source from the additional application, the inoculated *Rhizobium* also helps to increase and encourage growth and productivity by increasing the efficiency of nodulation and nitrogen fixation by soybean plants [24]. Thus, the plant gets a sufficient supply of nutrients, especially a supply of nitrogen which is very necessary at the vegetative stage. The number of leaves was also insignificant between 100% and 150% nutrient concentration. This means that

the soybean plant did not respond to additional nutrients to increase the number of leaves where only 100% nutrient concentration was sufficient to produce the maximum number of leaves in the early development stage of the two soybean varieties (Martina and Johanna). Furthermore, the number of leaves and the leaf area of the soybeans grew in a pattern that was almost similar when they were exposed to different nutrient concentrations. However, the leaf area increased significantly from 0% to 150% nutrient concentrations (Figure 8). It is the same as the findings reported by Haddad and Abahri [18] for the faba bean legume plant. They found that there was an increase in the leaf area with an increase in the nutrient solution concentration for all three varieties of tested faba beans. Since plants at higher nutrient concentrations allocated a larger fraction of carbohydrates to shoot growth than those at lower concentrations, plants at high nutrient concentrations could produce a wider leaf area. It was shown by Kang and van Iersel [25] through a study on salvia plants that there was a significant increasing trend in the plant leaf area ratio (LAR) at different nutrient concentrations (up to $1\times$ strength). The LAR is calculated based on the leaf area divided by the total dry weight. Because the plant LAR indicates how much leaf area a plant produces per gram of dry matter, a high LAR suggests that a plant is efficient at producing leaf area.

### 4.3. Nutrient Concentration and Variety Interacted on Plant Height

Nutrient concentration and variety have a significant interaction with plant height (Figure 2). The Martina variety was much higher than the Johanna at nutrient concentrations of 0% and 100%. Both varieties gave an almost similar response to the plant height when supplied with 50% nutrients. When the nutrient concentration was increased up to 150%, the Johanna variety was taller than the Martina where the difference was only 2 cm. This clearly shows that the determination of the nutrient concentration for a plant does not only depend on the type of plant but also the variety used because each variety has a different response to nutrients.

### 4.4. Variety and Week Number Interacted on Number of Leaves and SPAD Reading

Both the Martina and Johanna varieties also interacted with the week number on the number of leaves (Figure 5) and the SPAD reading (Figure 7). The interaction trend for both varieties on the number of leaves increased from Week 1 to Week 5. However, the Johanna variety produced a more significant number of leaves than the Martina variety in Week 1 and Week 4. In Week 2 and Week 3, the Johanna and Martina varieties had the same number of leaves. In Week 5, the leaves number of the Martina variety increased and was higher than the Johanna variety. As explained earlier, the growth of a plant in the early stage including the formation of leaves will increase with increasing time.

As for the SPAD reading, both the Martina and Johanna varieties showed an increase in the SPAD reading value only from Week 1 to Week 3 (Figure 7). It showed that the Johanna variety continued giving a higher SPAD reading than the Martina variety in Week 1, Week 2 and Week 3. Meanwhile, in Week 4 and Week 5, the SPAD reading for the Johanna variety decreased and was lower than for the Martina variety. The SPAD reading is leaf green color intensity which shows the chlorophyll content in leaves and stems. Plants use chlorophyll to produce food through photosynthesis [26,27]. The chlorophyll content is directly proportional to the rate of photosynthesis which increases from the youngest leaf to the mature leaf which can be described as "photosynthetically mature". After reaching a maximum value, chlorophyll content and photosynthesis rate decrease [28]. This is in line with the findings of this study which showed that the Johanna variety had a high photosynthesis rate in the early stages of plant growth and decreased in the last two weeks of its vegetative stage. However, this is contrast to the Martina variety which was more productive in producing food during the two weeks before its vegetative stage ended.

## 5. Conclusions

In conclusion, both the Martina and Johanna varieties grown in a controlled environment using the soilless substrate of expanded clay aggregate can grow well during

the early growth stages. The use of the nutrient solution of Advance Hydroponics of Holland at different rates influenced the plant growth such as the plant height, number of leaves and leaf area of the soybean varieties. The plant height of both the varieties was taller than the standard height that was determined in the previous study. In the first few weeks of growth, the Johanna variety produced more leaves and had a higher chlorophyll content than the Martina variety. The Martina variety, however, was the opposite. The leaf size of the Martina variety was larger than that of the Johanna variety. Both varieties required additional nutrients in the early growth stages between 100% and 150% nutrient concentrations based on their optimal growth at that rate. Early plant growth will influence reproductive and yield formation stages. The use of Advance Hydroponics of Holland as a nutrient source at the recommended rate is also expected to result in good growth at the reproductive stages and would produce high yields. Although the early growth patterns of the Martina and Johanna varieties were different, the grain yield would probably be comparable as they showed good early growth performance.

**Author Contributions:** Methodology, investigation, writing—original draft preparation, R.A.G.; Conceptualization, methodology, Á.T. and Z.K.; investigation, S.O. and N.K.; software, Z.K.; data curation, R.A.G., S.O. and N.K.; writing—review and editing, M.J.; supervision, M.J. and M.K.K.; funding acquisition, C.G. All authors have read and agreed to the published version of the manuscript.

**Funding:** This research was funded internally by the Hungarian University of Agriculture and Life Sciences. Malaysian Agricultural Research and Development Institute supported it.

**Institutional Review Board Statement:** Not applicable.

**Informed Consent Statement:** Not applicable.

**Data Availability Statement:** All data, tables and figures in this manuscript are original.

**Acknowledgments:** The authors would like to thank all the institute members of Agronomy for their assistance and support in this research. This research was supported by the Doctoral School of Plant Science, MATE. The students involved in this research were sponsored by the Malaysian Agricultural Research and Development Institute (MARDI) and the Stipendium Hungaricum respectively.

**Conflicts of Interest:** The authors declare no conflict of interest.

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
