# Peer review of "Soilless Culture Applications for Early Development of Soybean Crop (Glycine max L. Merr)"

_agriculture, doi:10.3390/agriculture13091713_

Round 1
Reviewer 1 Report
Soilless culture applications for early development of soy-bean crop is an interesting study and a beginning piece to better understand how soybean will thrive in the important reproductive stages and impacts on yield. This study can add valuable information to an area where little peer reviewed research has been done previously.
Please see the attached file for detailed information on the review but a few key points:
Throughout the paper chemical formulas need attention, spelling out on first use and using subscripts where appropriate
There is alot of redundancy of information provided in graphs and what is in the text. Some of the graphs show very little information and can be eliminated as the data is given in the text i.e. figure 4 and 10. There are also some sections of text that give the numerical value of what is shown in the graphs and is therefore not really necessary.
The conclusion section is lacking and needs to be improved instead of reiterating results explain what the results say about early growth performance in your study based on the metrics vs what high yielding soybean would be expected to show. Does adding nutrients at early growth stages make sense?
Citations throughout should be checked that they are formatted correctly.

Overall the paper is pretty well written. There are a few places where phrasing is awkward (noted in attached file) but a thorough read through should clear up most of the language usage issues.
Reviewer 2 Report
This is an interesting manuscript about the effect of nutrient concentration at 0, 50, 100, and 150% on the early development of two soybean varieties, Martina and Johanna, in a controlled environment and using the soilless culture as well as investigating the effects on plant height, number of soybean leaves, leaf area, and SPAD reading depending on the number of weeks and variety.
The present work was organized logically, and the results were reliable and persuasive. The results are well presented, their interpretation is relevant, and the methods are highly detailed. I would therefore recommend accepting this manuscript as a limitedly focused paper.
The English language of the manuscript needs to be revised in places.
Reviewer 3 Report
Line 44: CO2
Line 63: N2
Line 120: NH4-N
Line 183, 185: Table 1 and 2: NO3, NH4
Line 228, 267: Table 5: what is F?
Line 296: What is a SPAD? Tell me what the abbreviation is?
Line 309: What is F?
Line 253: Table 8: What is F (the same situation)
Line 393: Rhizobium sp. (if you want to use first time in article)
Round 2
Reviewer 1 Report
The revised version is an improvement on the original, the results section is much smoother and presents the data more clearly. The updated conclusions section also does a better job bringing together the main findings of the study. A couple of specific comments:
line 431: contra should be contrary
line 447: since this hasn't been tested yet it should perhaps be phrased "productive stages to potentially produce high yields"
line 449: again since this hasn't been tested but was touched on in the introduction perhaps rephrase "...good early growth performance, but more research is needed."
Any lingering issues should be easily resolved on the final read through.